# Signaling Pathways of Podocyte Injury in Diabetic Kidney Disease and the Effect of Sodium-Glucose Cotransporter 2 Inhibitors

**DOI:** 10.3390/cells11233913

**Published:** 2022-12-03

**Authors:** Xiutian Chen, Jiali Wang, Yongda Lin, Yiping Liu, Tianbiao Zhou

**Affiliations:** Department of Nephrology, The Second Affiliated Hospital, Shantou University Medical College, Shantou 515041, China

**Keywords:** SGLT2 inhibitors, diabetic kidney disease, podocyte, signaling pathways

## Abstract

Diabetic kidney disease (DKD) is one of the most important comorbidities for patients with diabetes, and its incidence has exceeded one tenth, with an increasing trend. Studies have shown that diabetes is associated with a decrease in the number of podocytes. Diabetes can induce apoptosis of podocytes through several apoptotic pathways or induce autophagy of podocytes through related pathways. At the same time, hyperglycemia can also directly lead to apoptosis of podocytes, and the related inflammatory reactions are all harmful to podocytes. Podocyte damage is often accompanied by the production of proteinuria and the progression of DKD. As a new therapeutic agent for diabetes, sodium-glucose cotransporter 2 inhibitors (SGLT2i) have been demonstrated to be effective in the treatment of diabetes and the improvement of terminal outcomes in many rodent experiments and clinical studies. At the same time, SGLT2i can also play a protective role in diabetes-induced podocyte injury by improving the expression of nephrotic protein defects and inhibiting podocyte cytoskeletal remodeling. Some studies have also shown that SGLT2i can play a role in inhibiting the apoptosis and autophagy of cells. However, there is no relevant study that clearly indicates whether SGLT2i can also play a role in the above pathways in podocytes. This review mainly summarizes the damage to podocyte structure and function in DKD patients and related signaling pathways, as well as the possible protective mechanism of SGLT2i on podocyte function.

## 1. Introduction

Diabetes is one kind of chronic metabolic disease, the most common of which is type 2 diabetes (T2D). In 2021, the global prevalence of diabetes among people aged 20–79 is estimated to be 10.5% (536.6 million people), and it will rise to 12.2% (783.2 million people) by 2045. In 2021, the global expenditure on diabetes-related health is estimated at 966 billion US dollars, which is expected to reach 1054 billion US dollars by 2045 [1]. Diabetes mellitus can shorten the life expectancy of patients [2]. The main reason for the rising incidence of diabetes is the aging population. The number of diabetic patients will continue to increase rapidly in the future. However, due to improved health care, effective prevention strategies may help to reduce the incidence of diabetes [3].

Diabetic kidney disease (DKD) is primarily caused by diabetes, while hyperglycemia and DKD are major risk factors for cardiovascular disease and overall mortality. Approximately 40% of patients with T2D have DKD [4]. DKD is a serious complication of diabetes and the main cause of renal failure, and there is currently a lack of effective treatment. Patients with DKD have a higher incidence of coronary artery disease, heart failure (HF), arrhythmias, and sudden cardiac death, and are much more likely to die from cardiovascular disease than from progression to end stage renal disease [5]. The incidence of diabetes-related complications has largely declined over the past two decades, however the incidence of DKD remains uncontrolled [6]. DKD speaks volumes about the growing role of diabetes as a major public health problem requiring more attention.

Podocytes are attached to the outside of the glomerular basement membrane (GBM), which together with vascular endothelial cells and GBM constitute the glomerular hemofiltration barrier. Podocyte damage disrupts the normal GBM structure, leading directly to proteinuria [7,8]. The experiment shows that diabetes is related to the decrease in podocyte number, hyperglycemia directly induces the apoptosis of cultured podocytes, and proteinuria increases with the decrease in podocyte number [9]. The number of podocytes in patients with DKD is decreased. Podocyte-specific NLRP3 inflammasome activation promotes DKD [10]. Moreover, DKD is closely related to autophagy of podocytes [11]. This suggests that DKD damages podocytes, while podocyte damage accelerates the progression of DKD.

Sodium-dependent glucose transporters 2 (SGLT-2) are the dominant transporters in sodium-dependent glucose transporters (SGLTs) that mediate the process of renal reabsorption of glucose. SGLT-2, mainly distributed in the S1 segment of the renal proximal convoluted tubule, is a transporter with low affinity and high transport capacity, and its main physiological function is to complete the reabsorption of 90% glucose in the glomerular filtration fluid in the renal proximal convoluted tubule [12,13]. Sodium-glucose cotransporter 2 inhibitors (SGLT2i) are a class of anti-hyperglycemic drugs approved for the treatment of T2D. These drugs block the reabsorption of glucose in the kidney by inhibiting SGLT2 thereby increasing urinary glucose excretion, promoting urination, and lowering blood glucose to improve glycemic control in a non-insulin-dependent manner. In addition to their ability to increase urinary glucose excretion and help to control blood glucose, SGLT2i have other properties that may be relevant for renal protection in DKD. SGLT2i play a renal protection role by inhibiting podocyte injury caused by diabetes. The drug can play a role through a variety of ways, including maintaining podocyte integrity, inhibiting podocyte apoptosis, enhancing podocyte autophagy, improving slit septum dysfunction, recovering podocyte epithelial-mesenchymal transition (EMT), and preventing podocyte loss.

This review mainly summarizes d the structural changes and functional damage of podocytes in patients with DKD, including its possible involvement in podocyte-related apoptotic autophagy, and discussed the role of SGLT2i in DKD’s involvement in regulating podocyte-related apoptosis and autophagy. The validity and safety of current animal models and clinical studies were summarized.

## 2. SGLT2i Provide Renal Protection in Animal Models

SGLT2i, such as the renal sodium-glucose transporter inhibitor T-1095 and TS-071 (Luseogliflozin), have a hypoglycemic effect when used in animal model experiments [14,15]. It also reduced proteinuria in animals [15] and delayed the progression of chronic kidney disease (CKD) [16]. SGLT2i showed significant recovery of renal cortex oxygenation and creatinine clearance [17,18]. SGLT2i can reduce blood pressure [19], and it can prevent the occurrence of angiotension II (Ang II)-dependent hypertensive renal fibrosis and Ang II-induced hypertensive renal injury (Table 1) [20,21].

It has been demonstrated in multiple animal experiments that SGLT2i can improve the terminal outcome of the kidney by promoting autophagy or improving related inflammation [22,23]. Similar effects on improving the inflammatory response were observed in models of infectious kidney injury and oxidative damage [24,25], which might be achieved by reducing the activity of NLRP3 inflammasomes [26,27]. In addition, autophagy may be improved through the mTOR1-related signaling pathway [28]. SGLT2i may alleviate the mitochondrial fission (Table 1) [29,30].

In metabolism, SGLT2i can improve the metabolism of animal models [31] and increase gluconeogenesis [32], possibly by inducing the expression of the key rate-limiting enzyme of gluconeogenesis [33]. In some animal models, SGLT2i can inhibit renal gluconeogenesis (Table 1) [34,35].

Furthermore, SGLT2i has no renal protection in progressive non-diabetic CKD models [36]. Notably, the use of high doses of SGLT2i also has the possibility of tumorigenesis (Table 1) [37].

## 3. SGLT2i Provide Renal Protection in Clinic

SGLT2i has shown significant benefits in clinical studies of patients with CKD. Many studies have shown that the use of SGLT2i can reduce the risk of severe cardiac and renal outcomes in patients [38,39,40,41], improve the cardiac and renal outcomes of patients, and effectively reduce the number of hospitalizations of patients [40,42], thereby reducing medical expenses in this area. The above studies have also shown that SGLT2i has significant effects on the reduction of glycosylated hemoglobin, type A1C (HbA1c), body weight, fasting blood glucose, and systolic blood pressure [43,44,45,46]. Many studies have shown that SGLT2i can significantly improve proteinuria, especially in patients with high proteinuria and thus improve the progress of urine albumin to creatinine ratio (UACR) in patients [47,48,49]. Although some studies have shown that the decrease in estimated glomerular filtration rate (eGFR) of this drug is not different from that of placebo [50], more studies have found that SGLT2i has a statistical significance in alleviating the decline slope of eGFR [51,52]. SGLT2i caused a significant increase in 24-h urine volume without an increase in urinary sodium when used in combination with a loop diuretic. The sodium benefits that SGLT2i may cause may be transient and only present early [53]. It is noteworthy that the study also found an early decrease in filtration rate (Table 2) [50,54,55].

In addition, hyperkalemia has a significant correlation with the terminal outcome of patients, and SGLT2i can also reduce the incidence of hyperkalemia in patients with CKD [56]. Combination of SGLT2i with angiotensin-converting enzyme inhibition can up-regulate the renin-angiotensin system in CKD (Table 2) [57,58].

Even though SGLT2i has beneficial effects on renal outcomes, the safety of SGLT2i is also important. Adverse events (AEs) associated with using SGLT2i include hypotension, dehydration, hypovolemia, syncope, urinary tract infection, genital infection, renal impairment, and so on. Most studies showed no significant differences in AEs compared to placebo [42,45,49]. Furthermore, some studies have even shown that SGLT2i reduces the incidence of AEs [59]. However, a few studies reported an increased incidence of glucosuria, urinary tract infections, and genital mycotic infections [41,46]. In addition, a transient decrease in eGFR in the early phase of SGLT2i treatment, followed by a gradual return to baseline levels, but SGLT2i still has an effect on the maintenance of eGFR after drug discontinuation [60], which also needs to be noted when administering SGLT2i. Since SGLT2i has the effect of lowering blood pressure and blood glucose, it is important to pay attention to the possibility of hypotension and hypoglycemia. In conclusion, SGLT2i appeared safe and effective in most clinical studies (Table 2).

## 4. Podocyte Injury-Related Signal Pathway

DKD is a serious complication caused by diabetes and occurs in 20–40% of people with diabetes [61]. DKD is related to significant podocyte damage. Diabetes also leads to podocyte apoptosis through a variety of apoptotic and autophagy mechanisms. Increased podocyte autophagy is associated with decreased mesangial dilatation, improved glomerular histology, and decreased proteinuria, finally, by inducing a shift from systemic glucose utilization to fatty acid oxidation [62]. The accumulation of lipids and toxic lipid metabolites in podocytes caused podocyte injury, and the accumulation of fatty acids in podocytes was related to the development of insulin resistance in vitro [63]. Mitochondrial oxidation of fatty acids stimulates the production of reactive oxygen species (ROS), further leading to tubular injury and apoptosis [64]. In addition, increased transport of albumin-bound fatty acids to the proximal tubules induced endoplasmic reticulum stress [6]. There are multiple pathways related to the regulation of apoptosis in podocytes. In the state of diabetes, each pathway is affected, and the common result is to promote the apoptosis of podocytes, destroy the basement membrane barrier formed by the participation of podocytes, and induce symptoms such as proteinuria, which in turn induces the further development of inflammation. The development of inflammation and DKD promote each other. The possible pathways of injury to podocytes by diabetes will be described below.

### 4.1. Phosphatidylinositol 3 Kinase (PI3K)/Protein Kinase B (AKT) Pathway

Diabetes weakens the PI3K/AKT pathway. The relative lack of insulin action and hyperglycemia lead to the impairment of Akt activity and Akt expression in type 2 diabetic mice [65]. The PI3K/AKT signaling pathway can inhibit the apoptosis of podocytes, and the activation of this pathway is an important part of maintaining the functional integrity of podocytes. CD2-associated protein (CD2AP) and p85 regulate the subunits of PI3K, absorb PI3K onto the plasma membrane and stimulate the PI3K-dependent AKT signaling pathway in podocytes. It can also promote puromycin aminonucleoside (PAN)-induced apoptosis of podocytes [66]. The related regulator miR-27a upstream of the pathway activates the pathway. MiR-27a up-regulates the activation of the PI3K/Akt signaling pathway at the protein and mRNA levels. Peroxisome proliferator-activated receptor (PPAR) inhibitors can inhibit PPAR-γ expression and increase AKT phosphorylation (Figure 1) [67].

In summary, type II diabetes weakens the activation of the PI3K/Akt pathway, and further weakens the inhibition of this pathway on podocyte apoptosis, leading to an increase in podocyte apoptosis, or inhibition of autophagy, resulting in kidney damage.

### 4.2. Mammalian Target of Rapamycin (mTOR) Pathway

One of the important kinases that regulate autophagy is mTOR. Studies have shown that the activity of mTORC1 is enhanced in diabetic animal podocytes. The overactivity of mTOR signal in podocytes plays a central role in the development of diabetic nephropathy models. Abnormal activation of mTORC1 leads to mispositioning of the slit diaphragm protein and induces EMT like phenotypic transformation in the presence of increased endoplasmic reticulum stress in the podocytes [68]. Studies on its related regulatory factors found that both thioredoxin interacting protein induced by hyperglycemia and mTOR can regulate cell autophagy [69,70]. PI3K/AKT is one of the pathways that regulates mTOR activity. Diabetes leads to abnormal activation of mTORC1 in podocytes, which in turn leads to insufficient autophagy in podocytes (Figure 1).

### 4.3. Janus Kinase (JAK)/Signal Transducer and Activator of Transcription (STAT) Pathway

Interleukin-6 (IL-6), an inflammatory mediator produced by diabetes, activates the JAK-signal transducer and activator of transcription 3 (STAT3) signaling pathway. IL-6 binds to cytokine receptors and induces dimerization of gp130 receptors [71]. Phosphorylated STAT3 (Tyr705) proteins interact to form homodimers, translocating to the nucleus, where they bind to specific DNA-responsive elements and regulate target gene expression (Figure 1) [72].

In human podocytes, advanced glycation end products (AGEs) induce p65 and STAT3 acetylation, while overexpression of p65 and STAT3 acetylation-deficient mutants inhibits AGEs-induced expression of NF-kB and STAT3 target genes [73]. The above studies have shown that activation of podocyte STAT3 by acetylation leads to an aggravation of nephropathy independent of changes in the upstream JAK signal (Figure 1).

The JAK/STAT pathway also inhibits podocyte autophagy. High glucose inhibits autophagy by activating the JAK/STAT pathway in mice and podocytes, thereby preventing the effective removal of damaged proteins and organelles from the body to prevent apoptosis, and finally aggravating podocyte injury and the progression of DKD (Figure 1) [74].

### 4.4. Transforming Growth Factor-β (TGF-β)/Smad Pathway

TGF-β activates mothers against decapentaplegic homolog (Smads). TGF is a key regulator of protein synthesis in the extracellular matrix (ECM) of renal cells. Increased expression of TGF-β mRNA and/or protein in podocytes of nephrotic patients [75]. TGF-β transmits signals through sequential activation of two cell surface receptors, serine-threonine kinase. In podocytes, TGF-β1 phosphorylates Smad2. The Smad2 and Smad3 proteins are activated by TGF-receptor kinases. Phosphorylated Smads form complexes with Smad4 and are then transferred to the nucleus, where they transduce signals to target genes (Figure 1) [76]. Smad7 inhibits signal transduction of NF-κB by cell survival factor, leading to TGF-β-mediated amplification of podocyte apoptosis [77]. TGF-β1 stimulates NOX4 through the activation of the Smad pathway by transcriptional podocyte-induced apoptosis [78]. TGF-β1-induced up-regulation of mitochondrial NOX4 through the TGF-β receptor-Smad2/3 pathway is the cause of ROS production, mitochondrial dysfunction, and apoptosis (Figure 1).

### 4.5. Wnt/β-Catenin Pathway

Up-regulation of the Wnt β-catenin signaling pathway was demonstrated in human DKD podocytes and streptozotocin (STZ)-induced diabetic mice [79]. Snail1, a direct downstream target of Wnt signal transduction, was up-regulated by β-catenin in DKD [80]. Ectopic expression of Wnt1 or stable β-catenin in vitro induces transcription factor Snail and inhibits nephrin expression, leading to podocyte dysfunction (Figure 1) [79].

Transient receptor potential cation channel, subclass C, member 6 (TRPC6) leads to proteinuria through a mechanism involving the activation of Wnt/β-catenin when mutated or chronically exposed to high glucose. High glucose induces apoptosis and differentiation of mouse podocytes, followed by a decrease in podocyte viability, leading to an increase in TRPC6 expression and activation of the Wnt/β-catenin pathway [81]. MiR-27a in DKD induced the deletion of podocyte-specific markers and increased apoptosis through PPARγ-mediated β-catenin activation, leading to podocyte injury and deterioration of renal function (Figure 1) [82].

### 4.6. MAPK Pathway

It has been found that TGF-β can induce apoptosis by activating mitogen-activated protein kinase (MAPK) p38 and the classical effector factor caspase-3 [77]. Diabetes-induced dual specificity phosphatase 4 (DUSP4) decreases and increases p38 and c-JunN-terminal kinase (JNK) activity, as well as induces podocyte dysfunction. Overexpression of DUSP4 inhibits high glucose exposure-induced p38, JNK, caspase 3/7 activity, and NOX4 expression [83]. At the same time, it has been found that the inhibitory protein kinase C-δ inhibits the decreased expression of DUSP4 and the activation of p38/JNK in podocytes and renal cortex of diabetic mice. Apoptotic signal-regulated kinase-1 (ASK1) is an upstream kinase of the p38 MAPK pathway, which promotes apoptosis once activated by inflammation and oxidative stress [84]. Evidence of increased ASK1 activity has been found in renal biopsy samples from patients with DKD (Figure 1) [85].

### 4.7. Inflammatory Reaction

Inflammation is one of the important mechanisms of kidney damage in DKD patients. Proteinuria in DKD patients promotes important changes of inflammatory state, endothelial dysfunction and coagulation-fibrinolysis balance [86]. Inflammation and oxidative stress are non-traditional risk factors (RF). The uremic toxin induces the production of free radicals in renal cells and induces an inflammatory response. In turn, the induced inflammation can be mediated by the production of ROS synthesis by macrophages [87]. There is a dynamic interaction between inflammatory response and oxidative stress, and these two processes are directly involved in renal cell injury [88]. Chronic inflammation and oxidative stress increase the incidence of DKD, which in turn promotes the development of inflammatory states [89].

### 4.8. NADPH Oxidase (NOX) Imbalance

Transgenic studies of overexpression of Nox-5 in rodents, especially in podocytes and mesangial cells, have demonstrated a possible pathogenic effect of this subtype [90]. Protein kinase C (PKC) has been proved to regulate the expression of NOX-2 and NOX-4 as well as ROS production [91]. The disorder of NOX leads to the increase of the NADH/NAD+ ratio and also leads to the activation of PKC, which exerts a pathogenic effect through the activation of TGF-β [92]. Relevant studies have shown that the pro-apoptotic effect of high glucose on podocytes is mediated by the activation of NOX, including ROS production, NF-kB and p38/MAPK [93,94].

### 4.9. NLRP3

The NLRP 3 inflammasomes are activated by damaging the cytoplasmic components of the associated molecular patterns, mitochondrial DNA, adenosine triphosphate (ATP), ROS, and glycoproteins to form active caspase-1 and active forms of IL-1 and IL-18. The activation of inflammasomes can also induce pyroptosis. The activation of NLRP3 inflammasomes aggravates podocyte autophagy and reduces the expression of podocyte nephrin, while the silence of NLRP3 effectively recovers podocyte autophagy and improves high glucose-induced podocyte injury. Autophagy plays a key role in maintaining lysosomal homeostasis in diabetic podocytes. Autophagy injury is involved in the pathogenesis of podocyte loss, leading to massive proteinuria in patients with DKD [95]. Serum factors associated with massive proteinuria in diabetes impair autophagy of podocytes but are not associated with diabetes itself. Podocyte autophagy deficiency may play a pathogenic role in the development of DKD into massive proteinuria [96].

### 4.10. Hyperglycemia Induces Apoptosis of Podocytes

In diabetes, the glomerulus enlarge beyond a certain limit and the podocytes are no longer able to maintain cellular proximity. In addition, compensatory podocyte hypertrophy is accompanied by substantial changes in podocyte morphology and protein expression. Such changes are associated with impaired function and susceptibility of podocytes to separate from the GBM, leading to a further decrease in podocyte density [97,98,99]. Hyperglycemia produces inflammation affecting podocyte nephropathy protein expression. In high glucose conditions, podocytes may attract macrophages by overexpressing the chemokine including vascular endothelial growth factor [100]. Macrophage-induced reductions in nephrin and podocin in cultured podocytes and isolated glomeruli [101]. This mechanism of injury involves tumor necrosis factor-α (TNF-α) produced by activated macrophages and has been shown to repress the nephrin gene at the transcriptional level [102]. Hyperglycemia produces AGEs, which bind to RAGE to activate FOXO4 transcription factors, which leads to podocyte apoptosis [103].

In summary, DKD can lead to podocyte apoptosis through multiple apoptosis-related pathways, including PI3K, JAK, TGF-β, Wnt, and MAPK. DKD can also lead to podocyte autophagy deficiency through mTORC pathway or other related pathways. Inflammatory factors, such as NOX, NLRP3, high glucose, changes in mechanical load environment and AGEs caused by diabetes, can also cause podocyte damage.

## 5. SGLT2i May Be Involved in the Regulation of the Podocyte Signaling Pathways

Dapagliflozin counteracts the effects of the PI3K/AKT/ Glycogen synthase kinase-3 beta (GSK-3β) pathway downstream of ROS [104]. Alpelisib is an α-selective phosphatidylinositol 3-kinase (PI3K) inhibitor, use of SGLT2i and a very low-carbohydrate diet in the absence of metformin are effective in reducing hyperglycemia during Alpelisib therapy [105]. SGT2/insulin-like growth factor-1 receptor (IGF1R)/PI3K signal transduction plays a key role in regulating the EMT of podocytes. Using SGLT2i in high-glucose model can significantly reduce the levels of SGLT2, IGF1R and phosphorylated PI3K, which indicates that SGLT2 inhibitor can inhibit the EMT of podocytes under diabetic conditions by downregulating IGF1R/PI3K pathway [106].

SGLT2i inhibits mTORC1 and prevents renal insufficiency [107]. Empagliflozin is shown to decrease IL-1β and TNF-α but significantly increase LC3-phosphatidylethanolamine conjugate.

LC3-I and bcl2/bax ratios, and its beneficial effects are activation of autophagy and inhibition of apoptosis. Empagliflozin attenuates renal injury in rats by promoting autophagy and mitochondrial biogenesis and by attenuating oxidative stress, inflammation and apoptosis [108].

In cardioprotective studies, dapagliflozin can exert cardioprotective effects by increasing EPO levels and activating P-Akt, P-JAK2 and pMAPK signaling cascades mediated by reducing apoptosis [109]. EPO was found to activate three pathways simultaneously: the Janus-activated cascade of kinase signal transduction and transcription activator (JAK2/STAT5), PI3K/Akt and extracellular signal-related kinase/MAPK (ERK/MAPK). In another cardio protection study, SGLT2i significantly increased STAT3 activity and STAT3 nuclear translocation [110].

SGLT2i inhibits TGF-β/Smad pathway activation. Dapagliflozin inhibits phosphorylation of the Smad3 junction (serine 204) induced by high D-glucose and TGF-β1 treatment, suggesting a mechanism of SGLT2-ERK-mediated TGF-β1/Smad3 signaling that induces pro-fibrotic growth factor secretion [111]. Another study showed that dapagliflozin could prevent fibroblast activation and mesenchymal transition through AMPKα-mediated inhibition of TGF-β/Smad signaling [112].

The expression of β-catenin in hepatocellular carcinoma cells is significantly down-regulated by carvedilol. Carvedilol promotes the proteasome degradation of β-catenin by increasing the phosphorylation of β-catenin [113]. The accumulation of β-catenin has been significantly inhibited by tofogliflozin. In tofogliflozin-treated mice, levels of blood glucose and mRNA expression of serum TNF-α, and pro-inflammatory markers in white adipose tissues were decreased, as was macrophage infiltration [114]. Dapagliflozin inhibits the inflammatory response by inhibiting the activation of the NF-κB pathway and TNF-α levels [104].

SGLT2i could inhibit the activation of the MAPK pathway. The protective effect of canagliflozin is associated with inhibition of p53, p38, and JNK activation. Canagliflozin enhances the activation of Akt and inhibited the mitochondrial pathway of apoptosis [115].

SGLT2i can reduce the activation of TGF-β in pig models and the expression of downstream Smad family [116], which promotes the improvement of renal fibrosis [117]. The simultaneous use of SGLT2i can improve the quality of life of patients [118]. The use of SGLT2i (empagliflozin) in patients with HFrEF effectively improved the VO_2_ peak, consistently improving both VO_2_ maximum and submaximal motor ability in the treated group [119]. Renal hypoxia occurs in patients with DKD, and the improvement of hypoxia with SGLT2i may also be another mechanism for its treatment of DKD. In addition, SGLT2i was able to reduce oxidative stress [117]. SGLT2i improves the energetics of tissues by shifting organ metabolism from glucose consumption to the use of free fatty acids and ketones [120] and thus improves ATP production [121]. SGLT2i can reduce podocyte lipid content, thereby promoting podocyte health and reducing proteinuria [62].

SGLT2i inhibits the inflammatory response in diabetes. Empagliflozin inhibits the up-regulation of NOX-2 and NOX-4 in the kidney of diabetic rats [122]. Empagliflozin weakens the activation of NLRP3 inflammasomes [123]. Dapagliflozin inhibits NLRP3 inflammasome activation and assembly, and subsequently pro-inflammatory IL [124]. The study by Leng et al. also showed that dapagliflozin could reduce ROS-NLRP3 activity [125]. Dapagliflozin reduced the inflammasome activation in diabetic mice (decreased mRNA levels of NALP3, apoptosis-associated speck-like protein containing a CARD (caspase recruitment domain), IL-1β, IL-6, caspase-1, and TNF-α). A similar conclusion has been reached in vitro models, indicating that the anti-inflammatory and anti-fibrosis effects may not be related to the hypoglycemic effect of SGLT2i [126].

Inhibition of SGLT improves renal cortical oxygen tension in diabetic rats, which may contribute to the improvement of tubular cell integrity and tubular albumin reabsorption [127]. In fructose-induced diabetes mellitus rats, Dapagliflozin down-regulated the expression of NADPH oxidase in RAGE-induced lens epithelial cells by inactivating glucose transporter and reducing ROS production [128]. Studies have shown that Dapagliflozin improves glucose toxicity by reducing the flow of elevated glucose into renal tubular epithelial cells under high glucose conditions. High glucose increases SGLT2 expression and glucose consumption, and produces AGEs. This eventually leads to excessive production of TGF-β1 and IL-8, as well as cell necrosis and apoptosis. Dapagliflozin improves the aggregation of AGEs and inflammatory response [129].

SGLT2i reduces glomerular pressure. The high expression of SGLT2 increases the reabsorption of sodium and glucose in proximal renal tubules, resulting in the decrease of sodium ions reaching the dense spots of distal renal tubules, which leads to the expansion of afferent arterioles. Therefore, the glomerulus shows high perfusion, high internal pressure and high filtration. The structural counterpart of renal ultrafiltration is renal hypertrophy, which is characterized by the increased volume of the glomerular cluster, Bowman’s cavity, renal tubular epithelium and renal tubular cavity at nephron level, resulting in the increase of GBM length and podocyte hypertrophy.

SGLT2i can reduce renal hyperfiltration, activate glomerular feedback, increase afferent arteriole tension, and reduce intraglomerular pressure through the mechanism of increased sodium secretion. SGLT2 inhibition has also been demonstrated to prevent renal hyperfiltration by reducing blood pressure and glomerular size and inhibiting renal growth factors [130].

## 6. No Associated Study Focusing on the Role of SGLT2i on Factors of Signal Pathway

More renal studies on the effects of SGLT2i on signaling pathways are expected. The above studies have shown that SGLT2i can regulate multiple pathways of PI3K, JAK, TGF-β, and MAPK in Figure 1 and affect cell apoptosis. Podocyte apoptosis is also regulated by the above pathways, so SGLT2i may play a role in alleviating diabetes-induced podocyte apoptosis. However, there is currently no related research that can explain whether the effects of SGLT2i on these pathways also play a role in podocytes, which may be a new research direction. At the same time, the research on the inhibition of Wnt/β-catenin pathway by SGLT2i did not involve its downstream Snail factor and TRPC6 channel that regulate podocyte apoptosis, so it is uncertain that SGLT2i has a regulatory effect on apoptosis caused by this pathway.

SGLT2i can directly prevent the down-regulation of autophagy mechanism in podocytes [131], and affect autophagy to cause podocyte changes.

However, the regulation of the above pathways by SGLT2i has demonstrated that the inhibitory effect of the drug on diabetes-induced cell damage caused by glycosuric nephropathy is achieved not only by inhibiting the urinary glucose in DKD but by regulating the gene expression through the pathway to inhibit the apoptosis of podocytes.

## 7. Conclusions

Studies on the damage mechanism of diabetes on the kidney, such as the variety of podocyte pro-apoptotic and anti-apoptotic mechanisms mentioned above, have shown that kidney podocyte damage caused by diabetes is the reason for the occurrence of proteinuria in diabetes. Therefore, in order to prevent diabetes from further developing into CKD, the intervention should be aimed at preventing or reducing podocyte apoptosis. As an anti-hyperglycemic drug, SGLT2i has been shown to improve the proteinuria in relevant diabetes treatment studies, maintaining the integrity of the podocytes, improving the slit septum dysfunction, recovering the EMT of the podocytes, inhibiting the apoptosis of the podocytes, enhancing the autophagy of the podocytes, preventing the loss of the podocytes, and other protective effects, thereby contributing to the reduction of proteinuria. Studies in other cells have shown that SGLT2i can regulate the related apoptotic pathways and inhibit cell apoptosis. However, in podocytes, there is no clear research on whether SGLT2i also plays a protective role in podocytes by regulating apoptosis. It is not clear whether SGLT2i can resist podocyte apoptosis and its anti-apoptosis mechanism. At present, this paper systematically describes the research of SGLT2i on intracellular apoptosis and autophagy-related pathways, which are the possible mechanisms of SGLT2i’s protective effect on podocytes. The description of the corresponding mechanism in this paper may provide another new idea for SGLT2i to study the protection mechanism of podocytes.

## Figures and Tables

**Figure 1 cells-11-03913-f001:**
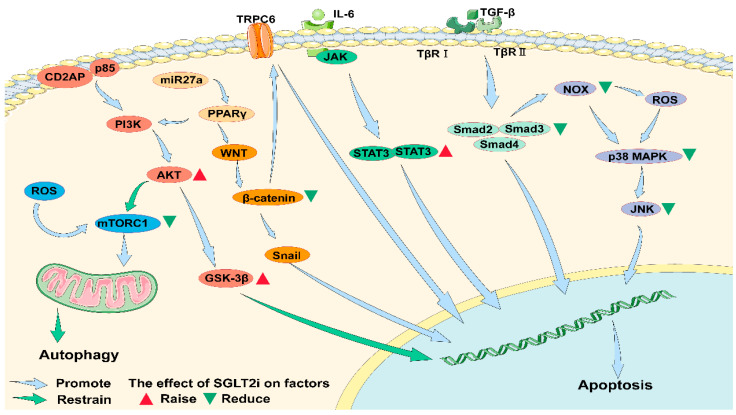
Mechanisms of promoting apoptosis and resisting apoptosis in podocytes. In DKD, the inflammatory response is enhanced in which ROS attenuate autophagy of podocytes via mTORC1. The combination of CD2AP and p85 promotes the activation of the PI3K/AKT pathway, which can lead to the apoptosis of podocytes. DKD inhibits the activity of the PI3K/AKT pathway and reduces its activity of inhibiting apoptosis. The Wnt/β-catenin pathway is activated in the diabetic state, which activates the downstream effector factor Snail and promotes the apoptosis of podocytes. At the same time, the activation of β-catenin is associated with the activation of the membrane channel TRPC6, which also promotes the apoptosis of foot cells. miR27a can activate the PI3K/AKT pathway and the Wnt/β-catenin pathway by mediating PPARγ. The increase of IL-6 in diabetes can activate the JAK/STAT3 pathway, and lead to apoptosis of podocytes. At the same time, the increased TGF-β can promote the activation of TβRI and TβRII and then promote the activity of Smad family. This pathway can also lead to podocyte autophagy. In addition, NOX can promote p38/MAPK pathway through ROS or directly, activate JNK downstream and promote podocyte autophagy. SGLT2i may exert effects on all the above pathways. Studies have shown that the use of SGLT2i can reduce the activity of mTORC1 in cells and promote autophagy. In addition, the use of SGLT2i can enhance the expression of AKT, reduce the expression of β-catenin and Smad families and inhibit the p38/MAPK/JNK pathway to inhibit apoptosis. However, studies have also shown that SGLT2i can promote the expression of STAT3 and thus promote apoptosis. ROS: reactive oxygen species; mTORC1: molecular target of rapamycin complex 1; CD2AP: CD2-associated protein; PI3K: phosphatidylinositol 3 kinase; AKT: protein kinase B; GSK-3β: glycogen synthase kinase-3 β; PPARγ: peroxisome proliferator-activated receptor; TRPC6: transient receptor potential cation channel, subclass C, member 6; IL-6: interleukin-6; JAK: janus kinase; STAT3: signal transducer and activator of transcription 3; TGF-β: transforming growth factor-beta; TβRI: the transformation growth factor-β type I receptor kinase domain; TβRII: the transformation growth factor-β type II receptor kinase domain; Smad: the drosophila mothers against decapentaplegic protein; NOX: nicotinamide adenine dinucleotide phosphate oxidase; MAPK: mitogen-activated protein kinase; JNK: c-JunN-terminal kinase.

**Table 1 cells-11-03913-t001:** SGLT2I-related animal experiments.

Author, Year	Medicines	Animal Species	Modeling Type	Research Results
Yamamoto et al., 2011 [14]	TS-071	Sprague-Dawley rats/Mice/Dog	STZ-induced diabetic rats and db/db mice	TS-071 has some advantages over current anti-diabetic drugs with the lower risk of pancreatic b-cells exhaustion and hypoglycemia
Adachi et al., 2000 [15]	T-1095	Male Wistar rats	STZ-induced diabetic rats	Plasma glucose, HbA1c, urinary albumin, kidney weight, and vacuolation of epithelial cells in the tubules in insulin-deficient diabetic STZ rats can be decreased by oral administration of T-1095
Motrapu et al., 2020 [16]	Empagliflozin	Mice	Uninephrectomized BKS-Lepr2/2 (db/db) mice treated with or without MRE served	Empagliflozin therapy significantly reduced albuminuria. In db/db 1K mice, Empagliflozin significantly reduced diffuse glomerulosclerosis in superficial as well as juxtamedullary nephrons
O’Neill et al., 2015 [17]	Phlorizin	Sprague-Dawley rats	Rat model of type 1 diabetes	The reduction in medullary PO_2_ in both control and diabetic kidneys, which results in medullary hypoxia. When SGLT is inhibited, renal cortex PO_2_ in the diabetic rat kidney is normalized
Rodriguez et al., 2015 [18]	Dapagliflozin	Han: Sprague-Dawley rats	Rat model of polystic kidney disease	DAPA-treated Cy/+ rats exhibited significantly higher clearances for creatinine and BUN and less albuminuria than vehicle-treated Cy/+ rats
Castoldi et al., 2021 [19]	Empagliflozin	Sprague Dawley rats	CsA (intraperitoneal injection) were administered for 4 weeks	Empagliflozin administration caused a reduction in blood pressure in CsA-treated rats and showed a protective effect on CsA nephropathy by decreasing renal fibrosis, type I and type IV collagen expression, macrophage infiltration and tyrosine hydroxylase expression
Castoldi et al., 2020 [20]	Empagliflozin	Sprague Dawley rats	Ang II osmotic minipumps	Prevent the development of renal fibrosis in Ang II-dependent hypertension
Miyata et al., 2021 [21]	Canagliflozin	Transgenic (Tg) mice	Agt/Cat-Tg mice were created by crossbreeding the Agt-Tgs and Cat-Tgs	There is a link between intrarenal RAS and SGLT2 expression and that SGLT2i ameliorates Ang II-induced renal injury independent of SBP
Jaikumkao et al., 2021 [22]	Dapagliflozin or Vildagliptin	Wistar rats	Rats were fed a high-fat diet for 16 weeks to induce obesity	The therapeutic effects of dapagliflozin attenuated pancreatic injury, pancreatic oxidative stress, endoplasmic reticulum stress, inflammation, apoptosis, and exerted renoprotective effects by restoring autophagic signaling in obese rats
Thongnak et al., 2022 [23]	Dapagliflozin and Atorvastatin	Wistar rats	Rats were HFF for 16 weeks and then treated with dapagliflozin with or without atorvastatin for 4 weeks	The combination therapy of dapagliflozin and atorvastatin has a positive effect on renal injury associated with autophagy and inflammasome activation induced by HFF in insulin-resistant rats
Maayah et al., 2021 [24]	Empagliflozin	Mice	LPS-induced inflammation or LPS-induced acute sepsis-induced renal injury	Empagliflozin significantly reduces systemic and renal inflammation to contribute to the improvements observed in an LPS-model of acute septic renal injury
Hasan et al., 2020 [25]	Canagliflozin	Rats	Isoprenaline (ISO)-induced renal oxidative damage in rats	Canagliflozin treatment of ISO-treated rats: AMP-activated protein kinase (AMPK), Akt, eNOS↑, iNOS, NADPH oxidase isoform 4 (NOX4)↓ Canagliflozin treatment improves kidney function in ISO-treated rats
Ye et al., 2022 [26]	Empagliflozin	Mice	Mice were fed an HFF (45% fat, 530 kcal/100 g)	Empagliflozin improves obesity-related kidney disease through reduction of NLRP3 inflammasome activity and upregulation of the HO-1–adiponectin axis
Ke et al., 2022 [27]	Dapagliflozin	Mice	Ischemia/reperfusion injury (I/R): block renal perfusion for 30 min	Dapagliflozin prevents NLRP3 inflammasome activation via promoting the mitochondrial tricarboxylic acid cycle metabolite itaconate
Tomita et al., 2020 [28]	Empagliflozin	Mice	db/db mice as a model of proteinuric DKD	Empagliflozin raised endogenous ketone body (KB) levels, KBs attenuated mTORC1-associated podocyte damage and proteinuria in diabetic db/db mice
Liu et al., 2020 [29]	Empagliflozin	Mice	Spontaneous diabetic KK-Ay mice	The Empagliflozin could alleviate apoptosis and oxidative stress in the kidney of KK-Ay mice. Empagliflozin may rescue the dephosphorylation of DRP1S637 and alleviate the mitochondrial fission via an AMPK dependent pathway
Otomo et al., 2020 [30]	Ipragliflozin or Insulin	Mice	Ins2Akita mice with KK/Ta background (KK/Ta-Akita mice)	Ipragliflozin treatment contributed to amelioration of proximal tubular protein overload, mitochondrial morphological abnormality, and renal oxidative stress and tubulointerstitial fibrosis.
Domon et al., 2021 [31]	Empagliflozin	Rats	Male enlarged kidney (DEK) rats	Treatment with empagliflozin: reduced blood glucose concentration, food intake, adeps renis, polyuria, polydipsia, urinary excretion of proteins, electrolytes↓, body weight↑. Empagliflozin could ameliorate systemic metabolism and improve renal tubule function in diabetic condition
Trnovska et al., 2021 [32]	Empagliflozin	Rats	Hereditary hypertriglyceridemic (hHTG) rats, a non-obese model of dyslipidemia, insulin resistance, and endothelial dysfunction	Empagliflozin: weight gain, fasting blood glucose, triglycerides, pyruvate, alanine↓, HDL-cholesterol, ketone bodies, leucine↑. In the liver, adipose tissue and kidney, empagliflozin up-regulated expression of genes involved in gluconeogenesis and down-regulated expression of genes involved in lipogenesis along with reduction of markers of inflammation, oxidative stress and cell senescence
Jia et al., 2018 [33]	Dapagliflozin	Mice	db/db mice	Dapagliflozin induced the expression of gluconeogenic key rate-limiting enzymes through increasing the expression levels of FoxO1 in the kidney and liver
Swe et al., 2020 [34]	Dapagliflozin	Wistar rats	HFF	Glucose tolerance was improved in obese prediabetic rats by suppressing renal glucose release from not only glucose reabsorption but also renal gluconeogenesis through improving renal cortical insulin signaling and oxidative stress
Vallon et al., 2014 [35]	Empagliflozin	Mice	Streptozotocin-diabetic mice, type 1 diabetic Akita mice	Empagliflozin attenuated/prevented the increase in SBP, glomerular size, and molecular markers of kidney growth, inflammation, and gluconeogenesis in Akita
Zhang et al., 2016 [36]	Dapagliflozin	Sprague-Dawley rats	5/6 (subtotally) nephrectomized rats, a model of progressive chronic kidney disease (CKD)	Glycosuria, hypertension, heavy proteinuria and declining GFR, glomerulosclerosis, tubulointerstitial fibrosis or overexpression of the profibrotic cytokine, transforming growth factor-ß1 mRNA were unaffected by dapagliflozin. SGLT2 inhibition does not have renoprotective effects in this classical model of progressive non-diabetic CKD
Knight et al., 2018 [37]	Empagliflozin	Mice	CD-1 mice	The development of renal tumors only in high-dose males (1000 mg)

STZ: streptozotocin; HbA1c: glycosylated hemoglobin, type A1C; CsA: cyclosporine-A; DAPA: dapagliflozin; Ang II: angiotensin II; RAS: renin-angiotensin system; SBP: systolic blood pressure; LPS: lipopolysaccharide; HFF: high-fat feed.

**Table 2 cells-11-03913-t002:** Clinical study of SGLT2i.

Author, Year	Medicines	Research Type	Race	Country	Patient’s Condition (CKD Stage, Proteinuria)	Change in End Indicators	Safety Evaluation
Bhatt et al., 2020 [38]	Sotagliflozin	Randomized controlled trial	White, Black, Asian	Multi-center	eGFR: 25 to 60 mL/min/1.73 m^2^	The risk of a composite of a sustained decline in the estimated GFR of at least 50%, end-stage kidney disease, death from renal or cardiovascular causes↓	Similar incidence of AEs with sotagliflozin and placebo
William G Herrington et al., 2022 [39]	Empagliflozin vs. Placebo	Randomized controlled trial	——	Multi-center	eGFR < 45 mL/min/73 m^2^: ≥45 to <90 ≥20 to <45 UACR: ≥200 mg/g	Empagliflozin therapy led to a lower risk of progression of kidney disease or death from cardiovascular causes than placebo	The rates of serious AEs were similar in the two groups
Perkovic et al., 2019 [40]	Canagliflozin (100 mg) vs. Placebo	Randomized controlled trial	White, Black, Asian, Other	Multi-center	eGFR of 30 to <90 mL/min/1.73 m^2^	The event rate of the primary composite outcome of end-stage kidney disease, doubling of the serum creatinine level, or renal or cardiovascular death was significantly lower in the canagliflozin group. The canagliflozin group also had a lower risk of cardiovascular death, myocardial infarction, or stroke and hospitalization for HF	The risk of kidney failure and cardiovascular events was lower in the canagliflozin group than in the placebo group. No significant differences in rates of amputation or fracture
Wanner et al., 2016 [41]	Empagliflozin	Retrospective analysis	Caucasian, African, Asian, etc.	Multi-center	eGFR < 30 (mL/min/1.73 m^2^)	Empagliflozin group: event or exacerbation of renal disease, risk of doubling serum creatinine levels, risk of renal replacement therapy↓	Genital infections↑
McMurray et al., 2021 [42]	Dapagliflozin	Retrospective analysis	White, Black or African American, Asian, Other	Multi-center	eGFR (25 to 75 mL/min/1.73 m^2^): ≥60 ≥45 to <59 ≥30 to <44 <30 UACR: 200–5000 mg/g	Dapagliflozin reduced the total number of HF hospitalizations (first and repeat) by 60%. Dapagliflozin reduced the overall slope of eGFR, which was similar in both HF and non-HF patients. Dapagliflozin was also effective in reducing the risk of kidney-specific renal lesions	Similar incidence of AEs with Dapagliflozin and placebo
Dagogo-Jack et al., 2021 [43]	Ertugliflozin	Randomized controlled trial	White, Black, Asian, Other	Multi-center	eGFR: 30 to 60 mL/min/1.73 m^2^	Ertugliflozin: HbA1c, body weight, SBP, the risk of HF↓, eGFR→	Similar incidence of UTIs with ertugliflozin and placebo
Heerspink et al., 2017 [44]	Canagliflozin vs. Glimepiride	Retrospective analysis	White, Black or African American, Asian, Other	Multi-center	UACR: ≥30 mg/g	Canagliflozin delayed the progression of renal disease, including eGFR and proteinuria, in type 2 diabetic patients within 2 years	Five patients experiencing acute renal failure or renal failure events, all in the canagliflozin group
Fioretto et al., 2018 [45]	Dapagliflozin	Randomized controlled trial	White, Black or African American, Indian/Alaska Native, Other	Multi-center	eGFR: 45 to 59 mL/min/1.73 m^2^	Dapagliflozin: HbA1c, body weight, fasting plasma glucose, SBP↓	Similar incidence of AEs with dapagliflozin and placebo
Allegretti et al., 2019 [46]	Bexagliflozin	Randomized controlled trial	White, Black or African American, Asian, Other	Multi-center	eGFR 30–60 mL/min/1.73 m^2^	Bexagliflozin: HbA1c levels, body weight, SBP, albuminuria↓	UTIs, genital mycotic infections↑
Jongs et al., 2021 [47]	Dapagliflozin	Retrospective analysis	White, Black or African American, Asian, Other	Multi-center	eGFR: 25 to 75 mL/min/1.73 m^2^ UACR: 200–5000 mg/g	Dapagliflozin significantly reduced albuminuria, with a larger relative reduction in patients with T2D. Among patients with UACR of 300 mg/g or greater at baseline, dapagliflozin increased the likelihood of regression in UACR stage	——
Jardine et al., 2021 [48]	Canagliflozin	Retrospective analysis	White, Black, Asian	Multi-center	UACR (mg/g) ≤1000 >1000–<3000 ≥3000 mg/g	Canagliflozin safely reduces kidney and cardiovascular events in people with T2D and severely increased albuminuria. In this population, the relative kidney benefits were consistent over a range of albuminuria levels, with greatest absolute kidney benefit in those with an UACR ≥ 3000 mg/g	——
Pollock et al., 2019 [49]	Dapagliflozin, Saxagliptin	Randomized controlled trial	White, Black, Asian, Other	Multi-center	UACR 30–3500 mg/g	Dapagliflozin and dapagliflozin–saxagliptin reduced UACR versus placebo	AEs or serious AEs were similar across groups
Kohan et al., 2016 [50]	Dapagliflozin	Randomized controlled trial	Caucasian, African, American, Asian, Other	Multi-center	eGFR (mL/min/1.73 m^2^): ≥90 ≥60 to <90 ≥30 to <60 <30	There was a small transient decrease in mean eGFR of dapagliflozin at Week 1, but it returned to near baseline values at Week 24 and remained stable at Week 102. At Week 102; the mean eGFR changes for dapagliflozin were not significantly different from placebo	Dapagliflozin was more likely than placebo to develop renal AEs in patients ≥65 years of age or CKD Phase 3
Heerspink et al., 2021 [51]	Dapagliflozin	Retrospective analysis	White, Black or African American, Asian, Other	Multi-center	eGFR: 25 to 75 mL/min/1.73 m^2^ UACR: 200–5000 mg/g	Dapagliflozin significantly slowed long-term eGFR decline in patients with chronic kidney disease compared with placebo. The mean difference in eGFR slope between patients treated with dapagliflozin versus placebo was greater in patients with T2D, higher HbA1c, and higher UACR	——
Cherney et al., 2021 [52]	Ertugliflozin	Prespecified exploratory analysis	White, Black, Asian, Other	Multi-center	eGFR (mL/min/1.73 m^2^): G1 ≥ 90 G2 ≥ 60 to <90 G3 < 60	Ertugliflozin has a favorable placebo-adjusted eGFR slope 0.75 mL/min per 1.73 m^2^ per year	——
Natalie A Mordi et al., 2020 [53]	Loop diuretic+Empagliflozin vs. Loop diuretic+Placebo	Randomized controlled trial	——	England	eGFR < 45 mL/min/1.73 m^2^	Empagliflozin caused a significant increase in 24-h urine volume without an increase in urinary sodium when used in combination with loop diuretic. The sodium benefit that Empagliflozin may cause may be transient and only present early	——
Heerspink et al., 2020 [54]	Dapagliflozin	Randomized controlled trial	White, Black, Asian	Multi-center	eGFR: 25 to 75 mL/min/1.73 m^2^ UACR 200–5000 mg/g	During the first 2 weeks, there was a greater reduction in the estimated GFR in the dapagliflozin group than in the placebo group. Thereafter, the annual change in the mean eGFR was smaller with dapagliflozin than with placebo	——
Oshima et al., 2020 [55]	Canagliflozin	Retrospective analysis	White, Black, Asian	Multi-center	eGFR: 30 to 90 mL/min/1.73 m^2^ UACR 300–3000 mg/g	Although acute drops in eGFR > 10% occurred in nearly half of all participants following initiation of canagliflozin the benefit of canagliflozin compared with placebo was observed regardless of the acute eGFR decline	——
Neuen et al., 2021 [56]	Canagliflozin	Randomized controlled trial	White, Asian, Black or African American, Other	Multi-center	eGFR (60–90 mL/min/1.73 m^2^): >60 45 to <60 <45 UACR: >300 mg/g ≤1000 1000 to <3000 ≥3000	Canagliflozin reduced the risk of hyperkalemia in patients with T2D and CKD compared with placebo, as well as patient use of potassium binders. There was no adverse effect on the occurrence of hypokalemia	A U-shaped association between serum potassium levels and renal and cardiovascular outcomes, such as an association between serum potassium levels < 4.0 or > 5.0 mmol/L and an increased risk of adverse outcomes
Antlanger et al., 2022 [57]	Empagliflozin	Randomized controlled trial and perspective study	——	Austria	eGFR: 15 to 59 mL/min/1.73 m^2^ UACR: 30 mg/g	Empagliflozin treatment resulted in a 1.5 to 2-fold increase in main RAS peptides in patients with diabetes compared with placebo. Compared with placebo, all main RAS peptides increased up to 100-fold more in the empagliflozin group, while plasma ACE activity and ACE2 levels remained suppressed	——
Sen et al., 2022 [58]	Dapagliflozin	Randomized controlled trial and retrospective analysis	White, Asian or of Middle Eastern	Multi-center	eGFR: ≥25 and ≤50 mL/min/1.73 m^2^ or eGFR: ≥25 mL/min/1.73 m^2^ UACR: 500–3500 mg/g	Compared to placebo, dapagliflozin increased plasma renin, aldosterone and copeptin levels	——
Heerspink et al., 2020 [59]	Dapagliflozin	Randomized controlled trial	White, Black, Asian, Other	Multi-center	eGFR: 25 to 75 mL/min/1.73 m^2^ UACR ≥ 200 mg/g	——	The rate of serious renal-related AEs was significantly lower in the dapagliflozin compared with the placebo group
Mayer et al., 2019 [60]	Empagliflozin	Randomized controlled trial	——	Multi-center	eGFR ≥ 30 mL/min/1.73 m^2^	Empagliflozin treatment in the EMPA-REG OUTCOME trial was associated with an initial reduction in eGFR from baseline to week 4 versus placebo (i.e., treatment initiation period). However, after week 4 until last value on treatment (i.e., the chronic treatment period), placebo-treated patients exhibited a significantly larger decline in eGFR than patients on empagliflozin did. After cessation of therapy, eGFR swiftly increased in empagliflozin treated patients versus those on placebo	——

eGFR: estimated glomerular filtration rate; HF: heart failure; HbA1c: glycosylated hemoglobin, type A1C; SBP: systolic blood pressure; UTIs: urinary tract infections; UACR: urinary albumin/creatinine ratio; RAS: renin-angiotensin system; ACE2: angiotensin-converting enzyme 2; T2D: type 2 diabetes. AEs: adverse events.

## Data Availability

Not applicable.

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
