# Peer review of "Signaling Pathways of Podocyte Injury in Diabetic Kidney Disease and the Effect of Sodium-Glucose Cotransporter 2 Inhibitors"

_cells, 2022, doi:10.3390/cells11233913_

Round 1

Reviewer 1 Report

Actually, this review is not complete, some parts are very poor developed, 12 pages of real manuscript, with 2/3 of each page occupied (as it is the MDPI draft) are not enough to describe the topic. Please see bellow my main suggestions regarding this manuscript:

1.     Last paragraph of the Introduction: please detail the novelty/special aspects that your study brings to the field. 

2.     Tables can be extended on the entire width of the page, as the MDPI draft allows it. They will have a better aspect, especially in the columns with many info/data.

3.     Please detail if SGLT2 can have potential beneficial effects related to podocyte function in non-diabetic patients given their beneficial effects proven in DAPA-CKD study. 

4.     In a separate chapter please better emphasis the beneficial role of SGLT-2 inhibitors as medications with anti-inflammatory action at the level of the kidney and how does inflammation affect podocyte function.  Also, detail in that chapter how inflammation and apoptosis specifically play a role in DKD. Please refer to the effects of finerenone, a new medication as an additional measure to improve podocyte function in CKD or DKD patients.  I suggest checking and referring to https://www.mdpi.com/2075-4418/11/8/1518   and https://doi.org/10.3390/medicina56030118

5.     Section 4 is not relevant at all. 4 lines cannot be considered as a section. Detail it much better or merge it to another section. Same for sections 6 to 8, and 10 to12. Why so many sections with so limited information?  The structure of the manuscript must be rethought in this regard.

6.     Figure 1. Each abbreviation used on the figure, must be detailed after the title of the figure, bellow it. As the Instructions for authors require, Abbreviations should be defined the first time they appear in each of 3 sections: the abstract; the main text; under the first figure or table. When defined for the first time, the acronym/abbreviation/initialism should be added in parentheses after the written-out form. Revise the entire manuscript in this regard.

7.     Last section must be Conclusions, not Conclusion, as there are more than 1 conclusion.

8.     Graphical part is very poor, only one figure. Please develop.

9.     References should be written in the MDPI style.

Author Response

Response to the reviewers

Dear reviewers,

We are hereby submitting our revised manuscript entitled “Signaling pathways of podocyte injury in diabetic kidney disease and the effect of sodium-glucose cotransporter 2 inhibitors” (Manuscript ID: cells-2016833). In the past several days, all the authors participated in revising this manuscript. We have fully addressed the comments of the reviewers. We hereby provide our point-by-point responses to all the concerns as detailed below.

Reviewer 1:

Comments and Suggestions for Authors

Actually, this review is not complete, some parts are very poor developed, 12 pages of real manuscript, with 2/3 of each page occupied (as it is the MDPI draft) are not enough to describe the topic. Please see bellow my main suggestions regarding this manuscript:

  1. Last paragraph of the Introduction: please detail the novelty/special aspects that your study brings to the field.

Response: Thank you very much for your suggestions! We have revised as “The purpose of this review is to summarize the renal protective effect of SGLT2i in DKD animal models and clinical studies, as well as its potential molecular mechanisms and signal pathways to protect podocytes”. Thank you very much again!

  1. Tables can be extended on the entire width of the page, as the MDPI draft allows it. They will have a better aspect, especially in the columns with many info/data.

Response: Thank you very much for your suggestions! The width of the table has been modified accordingly. We have revised them. Thanks again!

  1. Please detail if SGLT2 can have potential beneficial effects related to podocyte function in non-diabetic patients given their beneficial effects proven in DAPA-CKD study.

Response: Thank you very much for your suggestion! We try to add the content of this point. But we found it very difficult. DAPA-CKD is a clinical study, which only focuses on clinical indicators such as glomerular filtration rate and urine protein. The research of podocyte is the basic research of animal experiment. DAPA-CKD did not pay attention to the specific changes of podocytes. We understand that the opinion of the reviewer is to explain the protective effect of SGLT2i on podocytes in the non diabetic state. However, the topic we discussed in this paper is the renal protection of SGLT2i in DKD and its effect on podocytes. So, we can find a good place to mention it. Thank you very much again!

  1. In a separate chapter please better emphasis the beneficial role of SGLT-2 inhibitors as medications with anti-inflammatory action at the level of the kidney and how does inflammation affect podocyte function. Also, detail in that chapter how inflammation and apoptosis specifically play a role in DKD. Please refer to the effects of finerenone, a new medication as an additional measure to improve podocyte function in CKD or DKD patients.  I suggest checking and referring to https://www.mdpi.com/2075-4418/11/8/1518   and https://doi.org/10.3390/medicina56030118

Response: Thank you very much for your suggestion! There is another paragraph describing the effect of inflammation on the morphology and function of podocytes, and the effect of SGLT2i on inflammation is also mentioned in the follow-up. The suggested reference is also helpful for the article information. We have revised them. Thanks again!

  1. Section 4 is not relevant at all. 4 lines cannot be considered as a section. Detail it much better or merge it to another section. Same for sections 6 to 8, and 10 to12. Why so many sections with so limited information? The structure of the manuscript must be rethought in this regard.

Response: Thank you very much for your suggestion! Section 4, 6, 7, 8, 10, 11, 12 have been enriched. We have revised them. In order to make the content clear, we recoded the serial number. Thanks again!

  1. Figure 1. Each abbreviation used on the figure, must be detailed after the title of the figure, bellow it. As the Instructions for authors require, Abbreviations should be defined the first time they appear in each of 3 sections: the abstract; the main text; under the first figure or table. When defined for the first time, the acronym/abbreviation/initialism should be added in parentheses after the written-out form. Revise the entire manuscript in this regard.

Response: Thank you very much for your suggestions! The acronym/abbreviation/initialism have been added in parentheses after the written-out form. Each abbreviation used on the figure have detailed after the title of the figure. We have revised them. Thanks again!

  1. Last section must be Conclusions, not Conclusion, as there are more than 1 conclusion.

Response: Thank you very much for your suggestions! The corresponding content of the conclusion has been modified. We have revised them. Thanks again!

  1. Graphical part is very poor, only one figure. Please develop.

Response: Thank you very much for your suggestions! Note the content of the graph to supplement it accordingly, and explain the graph in words below the graph. We tried to add pictures, but found that Part 4 and Part 5 can be explained with a figure. We have revised the figure 1. Thanks again!

  1. References should be written in the MDPI style.

Response: Thank you very much for your suggestions! The style of the article has been revised accordingly. We have revised them. Thanks again!

All the authors have participated in revising this manuscript. We are aware that Cells is an outstanding journal, and thus we hope that our manuscript can be published in this journal. Thank you!

                                              Sincerely yours,

                                              Tianbiao Zhou

Reviewer 2 Report

The authors review the well-proven nephroprotective effects of SGLT2i and the molecular pathways responsible for podocyte protection.

            The authors are to be praised for an incredibly comprehensive review of the nephroprotective effects of SGLT2i both on animal models and on humans. Besides, the authors provide a summary of all the molecular pathways on whom SGLT2i exert benefits (PI3k/Akt, JAK/STAT, mTOR, TGF-b, Wnt, MAPK). The article offers interesting information, is updated, and reads well.

            This peer reviewer only offers the following comments:

-          The authors should mention the EMPA-Kidney study, whi has been presentd last week. Please summarize its main results (NEJM 2022 in press; doi: 10.1056/NEJMoa2204233)

-          The authors should mention that the natriuretic effect of SGLT2i is acute, but is mitigated (or disappears) in the mi-term (please quote Circulation. 2020 Nov 3;142(18):1713-1724).

-          When mentinoiong renal hypoxia in CKD, the authors should mention that SGLT2i improve oxygen consumption (). Therefore, SGLT2i improve the oxygenation of the kidney (J Am Coll Cardiol. 2021 Jan 26;77(3):243-255), which is another mechanism to explain the neproprotective benefits.

Author Response

Response to the reviewers

Dear reviewers,

We are hereby submitting our revised manuscript entitled “Signaling pathways of podocyte injury in diabetic kidney disease and the effect of sodium-glucose cotransporter 2 inhibitors” (Manuscript ID: cells-2016833). In the past several days, all the authors participated in revising this manuscript. We have fully addressed the comments of the reviewers. We hereby provide our point-by-point responses to all the concerns as detailed below.

Reviewer 2:

Comments and Suggestions for Authors

The authors review the well-proven nephroprotective effects of SGLT2i and the molecular pathways responsible for podocyte protection.

The authors are to be praised for an incredibly comprehensive review of the nephroprotective effects of SGLT2i both on animal models and on humans. Besides, the authors provide a summary of all the molecular pathways on whom SGLT2i exert benefits (PI3k/Akt, JAK/STAT, mTOR, TGF-b, Wnt, MAPK). The article offers interesting information, is updated, and reads well.

This peer reviewer only offers the following comments:

- The authors should mention the EMPA-Kidney study, whi has been presentd last week. Please summarize its main results (NEJM 2022 in press; doi: 10.1056/NEJMoa2204233)

Response: Thank you very much for your suggestion! We have added it in the table and in the text. Thanks again!  

- The authors should mention that the natriuretic effect of SGLT2i is acute, but is mitigated (or disappears) in the mi-term (please quote Circulation. 2020 Nov 3;142(18):1713-1724).

Response: Thank you very much for your suggestion! We have added it in the text. Thanks again! 

-When mentinoiong renal hypoxia in CKD, the authors should mention that SGLT2i improve oxygen consumption (). Therefore, SGLT2i improve the oxygenation of the kidney (J Am Coll Cardiol. 2021 Jan 26;77(3):243-255), which is another mechanism to explain the neproprotective benefits.

Response: Thank you very much for your suggestion! We have added it and added the related contents in the text. Thanks again! 

-The authors should mention that SGLT2i mitigate activation of TGF-beta in the porcine model (JACC Cardiovasc Imaging. 2021 Feb;14(2):393-407), which explains the improved tisular fibrosis (please reduce tisular fibrosis JACC Heart Fail. 2021 Aug;9(8):578-589)

SGLT2i improve quality of life (please quote Diabetes Metab Syndr. 2022 Feb;16(2):102417).

SGLT2i improve energetics of the tissue (Circulation. 2022 Sep 13;146(11):819-821) by switching organ metabolism away from glucose consumption towards the utilization of free fatty acids and ketones (J Am Coll Cardiol. 2019 Apr 23;73(15):1931-1944), which improve ATP production. SGLT2i reduce oxidative stress (please quote JACC Heart Fail. 2021 Aug;9(8):578-589)

Response: Thank you very much for your suggestion! The reference about oxygenation and metabolism has further enriched the content of this article and has been added to the main body of the article. We have revised them. Thanks again!

All the authors have participated in revising this manuscript. We are aware that Cells is an outstanding journal, and thus we hope that our manuscript can be published in this journal. Thank you!

                                              Sincerely yours,

                                              Tianbiao Zhou

Reviewer 3 Report

The authors tried to review the mechanisms of podocyte injury in diabetic-kidney disease and the effect of gliflozins.

The topic is very interesting. 

The reader can easily move from bench to bed side, even with typing errors.

The tables should be made more readable, probably arranged on a single, horizontal page.

Please improve the bibliography, quoting and commenting in the text the following papers:

Guo R, et al. SGLT2 inhibitors suppress epithelial-mesenchymal transition in podocytes under diabetic conditions via downregulating the IGF1R/PI3K pathway. Front Pharmacol. 2022 Sep 26;13:897167. doi: 10.3389/fphar.2022.897167. PMID: 36225569

DeFronzo RA, Reeves WB, Awad AS. Pathophysiology of diabetic kidney disease: impact of SGLT2 inhibitors. Nat Rev Nephrol. 2021 May;17(5):319-334. doi: 10.1038/s41581-021-00393-8. Epub 2021 Feb 5. PMID: 33547417

Author Response

Response to the reviewers

Dear reviewers,

We are hereby submitting our revised manuscript entitled “Signaling pathways of podocyte injury in diabetic kidney disease and the effect of sodium-glucose cotransporter 2 inhibitors” (Manuscript ID: cells-2016833). In the past several days, all the authors participated in revising this manuscript. We have fully addressed the comments of the reviewers. We hereby provide our point-by-point responses to all the concerns as detailed below.

Reviewer 3:

Comments and Suggestions for Authors

The authors tried to review the mechanisms of podocyte injury in diabetic-kidney disease and the effect of gliflozins.

The topic is very interesting.

Response: Thank you very much for your comment!

The reader can easily move from bench to bed side, even with typing errors.

Response: Thank you very much for your comment! We have corrected the full-text in the past days. Thanks again!

The tables should be made more readable, probably arranged on a single, horizontal page.

Response: Thank you very much for your suggestion! We have revised them. Thanks again!

Please improve the bibliography, quoting and commenting in the text the following papers:

Guo R, et al. SGLT2 inhibitors suppress epithelial-mesenchymal transition in podocytes under diabetic conditions via downregulating the IGF1R/PI3K pathway. Front Pharmacol. 2022 Sep 26;13:897167. doi: 10.3389/fphar.2022.897167. PMID: 36225569

DeFronzo RA, Reeves WB, Awad AS. Pathophysiology of diabetic kidney disease: impact of SGLT2 inhibitors. Nat Rev Nephrol. 2021 May;17(5):319-334. doi: 10.1038/s41581-021-00393-8. Epub 2021 Feb 5. PMID: 33547417

Response: Thank you very much for your suggestion! The above-mentioned reference has further supplemented the mechanism of SGLT2i in treating DKD in this article, and has been added to the text of the article. We have revised them. Thanks again!

All the authors have participated in revising this manuscript. We are aware that Cells is an outstanding journal, and thus we hope that our manuscript can be published in this journal. Thank you!

                                              Sincerely yours,

                                              Tianbiao Zhou

Round 2

Reviewer 1 Report

The authors responded to my requests.